# Evaluation of a New Rotator Cuff Trainer Based on Oscillating Hydraulic Damping

**DOI:** 10.3390/healthcare8010024

**Published:** 2020-01-28

**Authors:** Yinghao Wang, Jianfeng Wu, Hongchun Yang, Zhichuan Tang, Guozhong Chai

**Affiliations:** 1College of Mechanical Engineering, Zhejiang University of Technology, Hangzhou 310023, Zhejiang, China; wyhyk@126.com (Y.W.); chaigz@zjut.edu.cn (G.C.); 2Industrial Design and Research Institute, Zhejiang University of Technology, Hangzhou 310023, Zhejiang, China; yhc2016@zjut.edu.cn; 3College of Design, Zhejiang University of Technology, Hangzhou 310023, Zhejiang, China; ztang@zjut.edu.cn

**Keywords:** rotator cuff, shoulder rotation training, myoelectric test, sEMG, ergonomics evaluation

## Abstract

In order to provide a convenient way to strengthen the rotator cuff muscles and prevent rotator cuff injury, this study designed an innovative strength trainer specifically for shoulder rotator cuff based on oscillating hydraulic damping. We carried out a myoelectric testing experiment to evaluate the shoulder rotation training effect and compared the results with traditional training equipment to verify the feasibility and validity of the new rotator cuff trainer (RCT). Then, we further explored the influence of different training postures and motion speeds on shoulder rotation training. In the experiment, subjects used three types of equipment (RCT, dumbbells and elastic bands) to perform shoulder rotation training under two movement speeds and two motion postures. The surface electromyography (sEMG) signals of targeted muscles were collected in real time and then further analyzed. The experimental results showed that when using the RCT, the muscle force generation sequence was more aligned with the biomechanical principles of shoulder rotation than using the other two training methods, and the target training muscles had the higher percentage of muscle work. During RCT training, a higher speed of movement (120°/s) led to a higher degree of muscle activation; coronal axis rotation was better for the infraspinatus training, and sagittal axis rotation was better for teres minor training. Based on these results, the RCT was proved to be more effective than traditional training methods. In order to exercise the different muscles of rotator cuff more comprehensively and extensively, different postures should be selected. Furthermore, the movement speed can be appropriately increased within the safe range to improve muscle activation.

## 1. Introduction

Rotator cuff injury is one of the most common and frequent shoulder injuries, accounting for about 17–41% of shoulder joint injuries [1]. This injury can be caused by sudden sprains, internal deterioration of the tendon, or impaired shoulder stability caused by excessive abduction, manifesting as shoulder pain and limited mobility [2]. The rotator cuff, also known as the shoulder rotator cuff, consists of four muscles, including the supraspinatus, the infraspinatus, the scapula, and the teres minor muscle. It plays a key role in the stability of the shoulder joint and the smooth movement of the shoulder joint [3]. A previous study has found that resistance training for the rotator cuff muscles can enhance the muscle strength and tension of the rotator cuff muscles in the scapula plane, prevent tendon degeneration, and effectively improve the stability of the shoulder joint, which can effectively prevent rotator cuff injury and promote recovery of injuries [4].

The method of diagnosis, treatment and rehabilitation of rotator cuff injury have been widely studied [5,6]. In the course of therapy, isokinetic equipment is often used to test the range of movement of the shoulder joint and the muscle strength of rotator cuff (peak torque, relative peak torque, etc.) to judge the rehabilitation effect of rotator cuff injury [7,8]. However, isokinetic equipment is expensive, complex to operate and poor in generality [9], which is not suitable for ordinary fitness. At present, the academic research about strength training and prevention of rotator cuff injury is relatively few. Chen found that reinforcement exercises (the dumbbell side pulling in lateral position movement, the lateral dumbbell stretching movement, etc.) can improve the range of shoulder joint internal and external rotation and enhance the rotator cuff strength of internal and external rotation [10]. Wang et al. summarized 21 methods of resistance training for prevention of rotator cuff injury, including external rotation, internal rotation, abduction and adduction exercises under the standing position, supine position, and prone position [11]. Among the above methods, elastic band, thruster and dumbbell are the main training equipment for strength training to strengthen the rotator cuff. Elastic band is unable to quantify and adjust the resistance which is chosen according to subjective feeling the resistance is not constant and increases gradually with the increase of stretching range. Using a thruster or dumbbell is an isotonic training, but the muscles produce additional contraction against the gravity of weights (dumbbell). Because of the ever-present motion resistance when using the above training equipment, exercisers may be injured once reduce their muscle strength giving, which leads to some dangerous during training. The rotator cuff is a fragile muscle group and its safety is very important during training. Therefore, the safer equipment is necessary to develop for rotator cuff training.

In fluid flied, oscillatory hydraulic damping can produce a resistance opposite to the direction of rotation, and the resistance increases with an increase of the force, which almost will not generate inertia [12]. The motion characteristics of this hydraulic device is identical to shoulder joint movement. Based on the motion characteristics of oscillatory hydraulic damping and the movement patterns of the rotator cuff muscles, we designed a new oscillatory hydraulic damper and developed a new rotator cuff trainer (RCT) according to the height and size of Chinese men and women aged 18–65. The RCT realized the safer shoulder rotation training with a low cost. 

The aim of this study was to evaluate shoulder rotation training using the RCT. Dumbbells and elastic bands commonly used by trainers were also evaluated. By comparing the relevant muscle force generation sequence and the percentage of muscle work in the training process, the feasibility and validity of the RCT were explored and tested. The target muscle activation intensity analysis was used to explore the effect of different combinations of shoulder rotation posture and speed on the training effect. The proposed method is expected to provide more economical and safer training methods in national fitness and rehabilitation resistance training, and to prevent rotator cuff injuries for trainers.

### Design

Shoulder joint movement involves the sterno-lock joint, acromioclavicular joint, scapular joint, and glenohumeral joint and includes abduction and adduction, flexion and extension, external rotation and internal rotation, and circulation [13]. Related studies have pointed out that the external rotation and internal rotation of the shoulder joint can effectively exercise the muscles of the infraspinatus, scapularis, teres minor muscle, and deltoid muscle, and can improve rotator cuff function [14,15]. Therefore, training of the rotator cuff muscles is mainly carried out in the form of internal rotation and external rotation. 

The proposed RCT is shown in Figure 1. It is composed of five main parts: the oscillatory hydraulic damping device, angle adjusting mechanism, oscillatory arm, electric height adjustment support rod, and support base.

The core component of the rotary trainer is the oscillating hydraulic damping device, as shown in Figure 2. When the shoulder rotator is exercised, the shoulder, elbow joint, and the rotating shaft of the oscillatory hydraulic damping are all aligned in a straight line. The arm swings vigorously so that the swing arm drives the rotating shaft of the swing cylinder, and the rotating blade then rotates. The liquid in the corresponding cavity of the cylinder is pressed by the blade. The liquid flows through the throttle and enters the other cavity of the swing cylinder. Because the liquid is pressed against the blade to form a resistance opposite to the direction in which the arm swings, a resistance training effect can be produced. The exerciser can adjust the throttle area through the throttle valve to change the motion damping size. The angular velocity output of the hydraulic damping can be described by the following equation:(1)$$\omega =\frac{ca\sqrt{\frac{2}{\rho }\left(p-p_{0}\right)}}{R_{d}A}$$
where *ω* is the swing angular velocity, *c* is the throttle over current coefficient, *α* is the throttle area, *ρ* is the fluid density, *p* is the fluid pressure, *p*_0_ is the return pressure, *R_d_* is the oscillation cylinder equivalent radius, and *A* is the blade area [12]. From Equation (1), when the blade area and the liquid density are set to be constant, an increase in the force of the arm is manifested as an increase in the pressure of the pressure chamber, while the increase in the hydraulic pressure is not obvious. After the square root operation, the increase of the angular velocity is far less than the resistance increase. When the force of the arm decreased, the angular velocity decreased quickly. The damping device is safe for exercise because of no inertia [16]. Shoulder joint is an unstable active joint that it will be easy to be injured by a sudden external force [17]. Thus, it can be seen that no inertia is very important in strength training especially for rotator cuff. 

According to the size of different adults in China [18], we designed the electric support rod with height adjustable and the swing arm with the length adjustable on RCT. Before exercise, subjects can adjust the length of the swing arm and the height of the ground hydraulic damping device to make the trainer fit the ergonomic size for their own movement. The adjustment range of the swing arm length was 30–40 cm, and the height adjustment range of the oscillatory hydraulic damping device was 125–170 cm. In order to meet the need for shoulder rotation training of light resistance, medium resistance, and heavy resistance, a 12-gear knob was designed to set the exercise resistance. Each level of resistance is divided into four levels. Subjects with pre-existing rotator cuff or muscle damage can select the small resistance depending on individual condition. Moreover, the subject could also set the fixed angle of the RCT. The adjustment range was 0–90° [19], which can adapt to different shoulder rotation movements and exercise angles for the healthy or injured ones. After setting, the subject performed reciprocating shoulder rotation training.

The oscillating hydraulic damping device is a developed hydraulic device which is widely used in mechanical equipment. Compared with other large-scale fitness equipment, the manufacturing cost of the RCT is low and it can be used in gym and rehabilitation center in the future if it has good fitness effect. 

## 2. Materials and Methods

### 2.1. Subjects

In order to reduce the difference in physical variability, 20 healthy men without any skeletal muscle injuries at the shoulder joint were recruited. Their physical conditions were relatively close. The basic conditions were as follows: age 25 ± 2 years old, height 172.1 ± 4.3 cm, and weight 60.2 ± 3.2 kg. Three days before the experiment, all the subjects were required not to exercise vigorously to avoid muscle fatigue to avoid bias to the experimental results. The subjects were shown the correct use of the equipment prior to the experiment, and the subjects performed a period of practice adaptation and signed the informed consent form. The relevant experimental procedures were evaluated and approved by the Ethics Committee of Zhejiang University of Technology.

### 2.2. Apparatus

#### 2.2.1. Rotator Cuff Training Equipment RCT

##### RCT

The newly designed RCT was used as the first form of rotator cuff training equipment. The subject could turn the knob to adjust the motion resistance. The range of knob was from 1 to 12 gear.

##### Dumbbell

A set of Decathlon brand dumbbells was used as the second form of rotator cuff training equipment. The weight of the dumbbells was selected independently by each subject. The range of choice was from 1 kg to 5 kg.

##### Elastic Bands

A set of Decathlon elastic bands of different strengths was used as the third type of rotator cuff training equipment in this experiment, numbered from 1 to 5. The elastic band was stretched and deformed to produce resistance, but the resistance level cannot be quantified. Before the experiment, the elastic bands were pre-stretched by the subjects, and the elastic force was subjectively determined for subsequent experiment selection. 

#### 2.2.2. Data Acquisition Equipment

A multi-channel physiological signal acquisition system (MP150, BIOPAC Inc, GC, USA) was used in the experiment to collect EMG signal data for targeted muscle activity. A disposable Ag/AgCl electrode was used, and the gel-based bipolar electrode had a diameter of 30 mm and a distance between electrodes of 2 cm. The frequency of EMG data acquisition was set to 1000 Hz.

### 2.3. Experimental Procedure

#### 2.3.1. Warming up and Selection of Training Loads

Before the start of the experiment, the 20 subjects performed appropriate shoulder rotation warm-up exercises. After warming up, the subject used a dumbbell to do an external rotation of 90°. The maximum weight that could be lifted was 10.1 ± 0.7 kg. A dumbbell with a weight of 3 kg (up to 30% of the maximum weight) was then selected to perform an external rotation and a uniform rotation at an angular velocity of 60°/s, and the RPE value of the subject was tested by a subjective force RPE scale. Subsequently, with reference to the measured RPE values, different elastic bands and different gears of the RCT were used to perform the external and internal rotations at a uniform speed of 60°/s until the subject felt the output damping of the elastic band and the RCT was equivalent to that of a 3 kg dumbbell. We then recorded the label of the elastic band and the gear of the RCT. In the same way, the elastic band label and the RCT gear consistent with the damping of a 3 kg dumbbell were measured at a uniform speed of 120°/s. After testing, 12 subjects with similar subjective RPE values were selected to participate in subsequent EMG experiments.

#### 2.3.2. Selection of Target Muscles

In the rotator cuff muscle groups, the supraspinatus and scapular are double muscles, and surface EMG cannot be measured. In this experiment, the infraspinatus (IS) and teres minor (TM) were selected to represent the rotator cuff muscle group for subsequent evaluation. Also, several muscles with different functions related to the rotation of the shoulder were selected as the target muscles: specifically, anterior deltoid (AD), triceps brachii (TB), and extensor carpi ulnaris (ECU). The contraction of AD drives internal rotation of the shoulder joint [20]. The ECU is an auxiliary muscle in overcoming the resistance of rotation, and the TB helps to maintain the position in movement of shoulder rotation exercise [21]. The position of the muscle tested, and the position of the electrode are shown in Table 1.

#### 2.3.3. MVC Tests of Target Muscles

Firstly, the area of skin designated for electrode placement was cleaned with alcohol, and the electrode was placed onto the corresponding part of the five target muscles of the subject’s right arm. Then the maximum voluntary contraction (MVC) of the target muscle was measured; details of how measurements were obtained for each muscle can be found in previous studies on the IS muscle [22], TM muscle [23], AD [24], triceps [25], and ECU [26]. The MVC of each muscle was tested three times, and the maximum strength was maintained for 3 s in each test. The subject was given 5 min of rest after each MVC test.

#### 2.3.4. Experiment of Shoulder Rotation

After the end of all MVC tests, 20 min of rest was provided. After that, the subjects used three types of equipment: the RCT, dumbbell, and elastic band for muscle training. There were two training postures, namely the internal rotation and external rotation along the sagittal axis and the coronary axis, and the action range was 0−90°. The subjects performed the above two training actions at a uniform speed of 60°/s and 120°/s by means of metronome guidance. One task was training with an action at a rotational speed using one device, so each subject performed 12 tasks (3 devices × 2 actions × 2 angular velocities). Each task was performed in 10 groups. In order to avoid generating fatigue, a 3-min break was provided between two tasks. The experimental process is shown in Figure 3.

### 2.4. Data Processing

There were two comparisons in the experiment. The first one was to use the sEMG muscle force generation sequence and the muscle work percentage to evaluate the subject’s use of the RCT, the dumbbell, and the elastic band under the same exercise load. The purpose was to compare the training effects of the rotator cuffs of the three devices. The second was to use the muscle activation intensity index of sEMG to explore the effect of different postures and speeds on the training effect of the rotator cuff muscles when using the RCT. The purpose was to find better and more effective training movements and motion speed.

All sEMG signals of five target muscles during the experiment were collected through five signal channels. The AcqKnowledge software was used to evenly take points on the acquired sEMG curves at 0.01 second intervals to generate a new sEMG curve. The three sEMG evaluation indicators described in Section 2.4 were calculated. An sEMG picture of a complete exercise cycle was taken from each task of the subject, and the muscle activation sequence was sequentially recorded according to the change in the target muscle EMG signal over time.

The muscle force generation sequence refers to the activation sequence of each participating muscle when an inactive muscle is under the control of the central nervous system [27]. The muscle force generation sequence can be used to accurately analyze whether the subject’s movement is proper during shoulder rotation. In this experiment, it was mainly determined by observing the change in sEMG in the exercise cycle: the sEMG of a complete exercise cycle was taken from each task of the subject, and the muscle activation sequence was recorded according to the percentage of total muscle activation time.

The percentage of muscle work refers to the integral electromyogram (iEMG) of a muscle during the completion of a movement cycle as a percentage of the total integrated myoelectric signal of all muscles involved in completing the exercise [28]. It can reflect the importance of the muscle in completing the movement. The iEMG of the target muscle in each task was calculated as follows: (2)$$iEMG=\frac{1}{T}\int_{t}^{t+T}\left|EMG\left(t\right)\right|dt$$
where *t* represents the experimental data sample start time, and *t* + *T* is the sample end time.

According to the definition of muscle work percentage, the equation is as follows:
(3)$$W(k)=\frac{{iEMG}_{k}}{{iEMG}_{1}+{iEMG}_{2}+\ldots +{iEMG}_{n}}\times 100\%$$
where *k* is the target muscle, and *n* is the *n*th muscle, *W* is the percentage of work done by target muscle.

Muscle activation intensity is the mean value of muscle activation during the exercise cycle. In this experiment, the RMS value of EMG was used to express the muscle activation intensity [29]. The RMS of the target muscle in each task was calculated as follows: (4)$$RMS=\sqrt{\frac{1}{N}\int_{t}^{t+T}EMG^{2}\left(t\right)\cdot dt}$$
where *t* represents the experimental data sample start time, and *t* + *T* is the sample end time. Because there were large differences in the RMS values among different individuals, it could not be directly used to compare the intensity of muscle activation between individuals. In this study, MVC was used as a benchmark, and the equation for calculating muscle activation intensity was as follows: (5)$$RMS{\%}_{MVC}=\frac{RMS_{measured}}{RMS_{MVC}}\times 100\%$$

Statistical analysis of the processed data was performed using SPSS 19.0. 

## 3. Results

### 3.1. Muscle Force Generation Sequence

Figure 4, Figure 5 and Figure 6 show the percentage of initial time of each target muscle’s mobilization in an exercise cycle when using different exercise equipment for the shoulder rotation movement. It can be seen from the figure that when using the RCT, the order of mobilization of the five target muscles tested was: IS, TM, AD, ECU, and then TB. When training with dumbbells, the order of muscle mobilization was ECU, AD, TB, IS, and then TM. When using elastic band training, the order of muscle mobilization was: AD, TB, IS, TM, and then ECU. The order of IS, TM, and ECU varies greatly when training on three types of equipment. When internal rotations around the sagittal axis and the coronal axis were performed at different speeds on the same training equipment, there was no significant difference in the order of the measured muscles.

### 3.2. The Percentage of Muscle Work

Figure 7, Figure 8 and Figure 9 illustrate the percentage of total work done by all the muscles during a single exercise cycle when using different equipment for the shoulder rotation movement. Based on Figure 7**,** when using the RCT, the IS had the highest percentage of work, followed by TM, while TB had the least work (less than 10%). Compared with the movement speed of 60°/s, when the subjects exercised at 120°/s, the proportion of work done by IS increased significantly, while the proportion of work done by TM decreased. The difference for the other three target muscles was not evident. When the shoulder was rotated around the sagittal axis, the percentage of work done by IS was lower than that when the shoulder was rotated around the coronal axis. When the shoulder was rotated around the coronal axis, the percentage of work done by AD, TB and ECU was slightly lower than when the shoulder was rotated around the sagittal axis. When rotation around the coronal axis was performed at 60°/s, the percentage of work done by TM was higher than when rotation was around the sagittal axis, and there was a significant difference by t-test (*p* < 0.05). When the movement speed was 120°/s, the percentage of TM work in the sagittal axis rotation of the shoulder movement was higher than that of the coronary axis rotation, and there was a significant difference by the t-test (*p* < 0.05). Based on Figure 8 and Figure 9 IS was mainly used during the dumbbell task to perform the shoulder rotation movement, and IS had the highest proportion of work, followed by ECU. When the elastic belt was used for the shoulder rotation movement, the ECU had the highest work ratio, and the TM was the second.

Comparing the percentage of work data of the measured muscles using the three types of equipment, the percentage of work done by IS was the highest with the RCT, the second with the dumbbell, and the lowest with the elastic band. The percentage of work done by TM was the highest with the RCT, and similar with the dumbbells and elastic bands. TB had the lowest proportion of work done among all muscles regardless of the device used, but the percentage of work done by TB was even lower when the RCT was used as compared with the other two devices.

### 3.3. Muscle Activation Intensity

Through the muscle force generation sequence and the percentage of muscle work, it was found that the use of the RCT for shoulder rotation movement was superior to dumbbells and elastic bands and was more suitable for the training of target muscles (IS and TM). Further, the effects of different postures and speeds on the training effect of the rotator cuff muscles were evaluated using the RCT. Figure 10 shows the activation intensity of each target muscle. When the shoulder rotation movement was performed at a speed of 120°/s, the activation intensity of the muscle was significantly higher than at 60°/s, that is, the faster the speed of the shoulder rotation movement, the higher the muscle activation intensity. At both speeds, the IS activation intensity of rotation around the coronal axis was higher than that around the sagittal axis. The TM activation intensity in the coronal axis posture was lower than in the sagittal axis posture. MANOVA (2 posture × 2 speed) was used for each muscle to analyze the effect of posture and velocity. The statistical results showed that for AD, IS and TM, posture (F = 3.475, *p* < 0.05; F = 6.632, *p* < 0.05; F = 4.324, *p* < 0.05) and velocity (F = 7.134, *p* < 0.05; F = 6.243, *p* < 0.05; F = 5.411, *p* < 0.05) had significant effects on muscle activation intensity. For the TB and ECU muscles, posture (F = 0.432, *p* > 0.05; F = 0.231, *P* > 0.05; F = 0.134, *p* > 0.05) had no significant effect on muscle activation intensity, and velocity (F = 2.014, *p* < 0.05; F = 3.339, *p* < 0.05) had a significant effect on muscle activation intensity.

## 4. Discussion

### 4.1. Discussion on the Sequence Analysis of Muscle Force Generation

As each subject performed a specified action, the order of muscle force generation was recorded, which can be used to analyze whether the subject’s movement during the shoulder rotation movement was proper. The results of the muscle force generation sequence showed that when using the three types of equipment, there was a significant difference in the mobilization order of the muscles, and the order of the two exercise postures of the same equipment was largely the same. In a group of experimental actions, the subject completed external rotation and then completed internal rotation. Therefore, the results showed here were for the case of external rotation. According to the biomechanical analysis, during the external rotation, the contraction of IS, TM, and teres major muscle causes the tibia to rotate. After that, the deltoid muscle plays a synergistic force to maintain the stability of the shoulder. Finally, the arm muscles are driven to make the humerus and the ulnar rotate. Previous studies have found that in processes such as volleyball players smashing, badminton players killing the ball, boxing athletes punching, mobilization starts in the large muscle group: the pectoralis major and latissimus dorsi muscles first activate, driving the rotator cuff muscles and the forearm muscles, which drive the arm muscles to complete complex upper limb movements [30,31,32]. In the current study, when the subjects used the RCT to do the training, the order of mobilization of the five target muscles was: IS, TM, AD, ECU, and then TB, which was more in line with the biomechanical principles of shoulder rotation compared with the use of dumbbells and elastic bands.

According to the analysis of the different force sequences displayed by the experimental results, we found that the subjects need to overcome the weight of the dumbbells and certain arm gravity when training with dumbbells. Therefore, at the beginning of the exercise, the ECU, AD, and TB of the subjects took the lead in order to maintain the posture and stability of exercise.

When using the elastic band training, the subject only needs to overcome the arm gravity to maintain the posture, and the weight required to overcome the gravity is smaller than the dumbbell training. Therefore, the AD is the first to exert force. In the RCT, because the equipment is equipped with elbow support, the subject does not need to overcome any gravity during the whole movement, and the damping of the shoulder is always opposite to the direction of motion. Therefore, the order of muscle mobilization during the movement is more in line with the biomechanical principles of shoulder rotation. This ensures the continuous movement process of the shoulder rotation training process, which in turn produces the best force effect, keeping the shoulder rotation movement smooth and uninterrupted. What’s more, the elbow support will help the patients with pre-existing rotator cuff or muscle damage to complete the training in a safe way, because the muscles may not have enough strength to overcome the gravity of their arm [33].

### 4.2. Discussion on the Sequence Percentage of Muscle Work

The percentage of muscle work done refers to the activity contribution rate of the muscle during the completion of the specified action, reflecting the degree of participation of the target muscle in the exercise. According to the experimental results, among the three types of equipment, the work of IS and TM in the training of the RCT has the highest proportion, which is much higher than that of the dumbbells and elastic bands, while the work of AD, TB and ECU has the highest proportion when training with dumbbells or elastic bands. The ratio is significantly higher than the RCT.

Based on the experimental results, further analysis was carried out on the actions and forces when using the three training devices. In the experiment, a set of shoulder training consisted of two actions: external rotation and internal rotation. When training with an RCT, whether performing external rotation or internal rotation, the oscillatory hydraulic damping always produces the opposite resistance to the movement direction during the movement. When the movement speed remains the same, the generated damping remains the same [12]. Therefore, the RCT can enable the subject’s rotator cuff muscle group to continuously and effectively exercise during both external rotation and internal rotation.

When the subject used the dumbbell to perform the external rotation, the gravity of the dumbbell was always vertically downward. At the beginning of the movement, the direction of the motion was opposite to the direction of gravity, and the rotation resistance of the rotator cuff muscle group was the greatest. As the external rotation angle increased, the resistance in the direction of rotation gradually decreased. In the late stage of external rotation, the subjects were balanced by the muscles of the ulnar wrist extensor, biceps, triceps, and deltoid muscles [34]. Internal rotation is the reverse action of the external rotation, and the process of the muscle group is opposite to external rotation. In the whole set of movements, the rotational damping that can effectively stimulate the power of the rotator cuff changes from large to small and then larger, and it is always smaller than the weight of the dumbbell. Therefore, the rotator cuff muscles have relatively small effective work to overcome the rotational resistance. Different from the training process of dumbbell training, when the subject uses the elastic band to do a set of shoulder-rotation action, the elastic band is deformed such that the output elastic force becomes smaller and larger and then smaller. Similarly, because the other end of the elastic band is fixed, the elastic force generated by the elastic band during the rotation of the shoulder is not in the opposite direction to the movement. Part of the elasticity requires the muscles of the ulnar wrist extensor, biceps, triceps, deltoid muscles to overcome the balance to maintain the balance of exercise, so the rotator cuff muscles are relatively less effective.

From the Figure 7, Figure 8 and Figure 9, we also found that different speeds have inconsistent effects on different muscles. IS is one of the most powerful rotator muscles of the shoulder, and TM is an auxiliary muscle for shoulder rotation [35]. When using the RCT, with the increase of the speed, the IS force output increases quickly to maintain a faster speed. Unlike IS, the increase of TM force output is far less than rotator cuff muscles. As a result, the percentage of muscle work of IS increased with a higher speed, and the percentage of muscle work of TM decreased. When using dumbbell or elastic band, the AD and TB contributed a lot to keep the position of movement which led to a better work for AD and TB muscles than RCT. All tested muscles might be almost equally important to the shoulder exercise that the percentage of muscle work had no significant change. 

Based on experimental results and kinematic analysis, the use of the RCT can provide effective rotational damping. During exercise, the ineffective work is relatively small, and the RCT is more suitable than dumbbells and elastic bands to specifically strengthen the rotator cuff muscle. What’s more, a higher speed will be better to IS building.

### 4.3. Discussion on the Muscle Activation Intensity

In this experiment, the RMS index of the surface EMG signal is used to express the muscle activation intensity, and the RMS is the change in the amplitude of the target muscle activation, that is, the effective discharge value. According to experimental statistics, for all five muscles, speed has a significant effect on muscle activation intensity. Previous studies have pointed out that the size of RMS is determined by the combination of muscle strength and discharge in actual exercise [36]. As the rhythm of the movement increases, the muscle contraction level increases, the amount of discharge increases, and the muscle force output increases to maintain a faster movement speed. Therefore, in the experiment, when the shoulder rotation training is performed at a higher speed (120°/s), the activation intensity of the muscle is stronger. Compared with the activation degree of target training muscles (IS and TM), it is found that when the shoulder rotation is performed at two speeds, the IS activation intensity of the coronal rotation posture is higher than that of the sagittal rotation posture. The TM activation intensity of the coronary rotation posture is lower than that of the sagittal axis rotation posture. For AD, IS, and TM muscles, posture has a significant effect on muscle activation intensity. For the other two muscles, posture has no significant effect on muscle activation intensity. Coronal axis rotation is better for the IS training, while the sagittal axis rotation is better for TM training. The trainer can select different exercise postures for different muscles in the rotator cuff muscle group to achieve the purpose of comprehensively exercise.

According to the above analysis, when using the RCT, it is recommended to use all training postures to stimulate the different rotator cuff muscles to achieve the effect of comprehensive exercise. In the safe range of muscle strength training, the trainer can appropriately accelerate the rhythm of the exercise according to his or her own ability to enhance the activation degree of the rotator cuff muscles, thereby achieving better training results.

## 5. Conclusions and Future Prospects

In this paper, based on the oscillatory hydraulic damper, an RCT for the rotator cuff muscles is designed, and a safer shoulder training is realized at a low cost. Through ergonomic experiments, three types of equipment, the RCT, dumbbells, and elastic bands, are compared. The order of the muscle mobilization when the subjects use the RCT is most in line with the biomechanical principles of the shoulder-rotation motion, and ineffective work is relatively small, which is suitable for trainers to carry out intensive training of rotator cuff muscles. In addition, when using the RCT, it is recommended that the trainer appropriately accelerates the rhythm of the movement within the safe range according to his/her own ability, and at the same time switch between different training postures to more comprehensively and deeply enhance the muscle strength of different rotator cuff muscle groups. In future studies, we will add the test for female and patients, to verify the feasibility and validity of RCT to different people. And then increase the test of deep rotator cuff muscles (scapular and supraspinatus muscles) to explore the effect of different postures and speeds on the absolute strength, endurance and explosive force of rotator cuff muscles. Then, according to different individuals, the personalized exercise prescription of rotator cuff strength training can be established.

## Figures and Tables

**Figure 1 healthcare-08-00024-f001:**
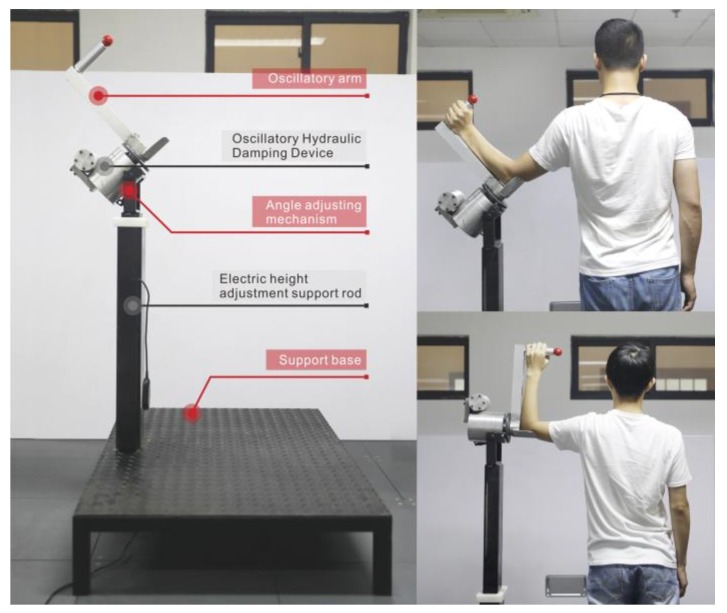
Pictures of the RCT and its usage.

**Figure 2 healthcare-08-00024-f002:**
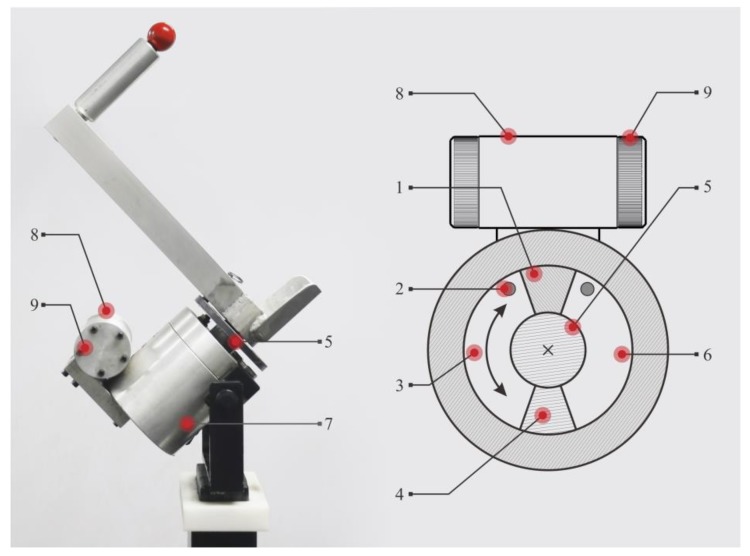
Oscillatory hydraulic damping device. 1—Fixed blade; 2—Rotating blade; 3—Rotating axis 4—Cavity 1; 5—Cavity 2; 6—Throttle; 7—Hydraulic cylinder; 8—Throttle valve; 9—Throttle controlling knob. Pictures of the experimental progress.

**Figure 3 healthcare-08-00024-f003:**
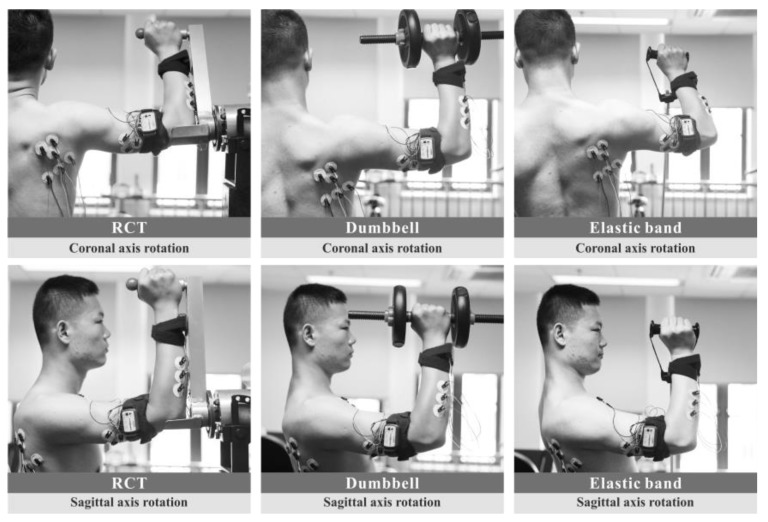
Pictures of the experimental progress.

**Figure 4 healthcare-08-00024-f004:**
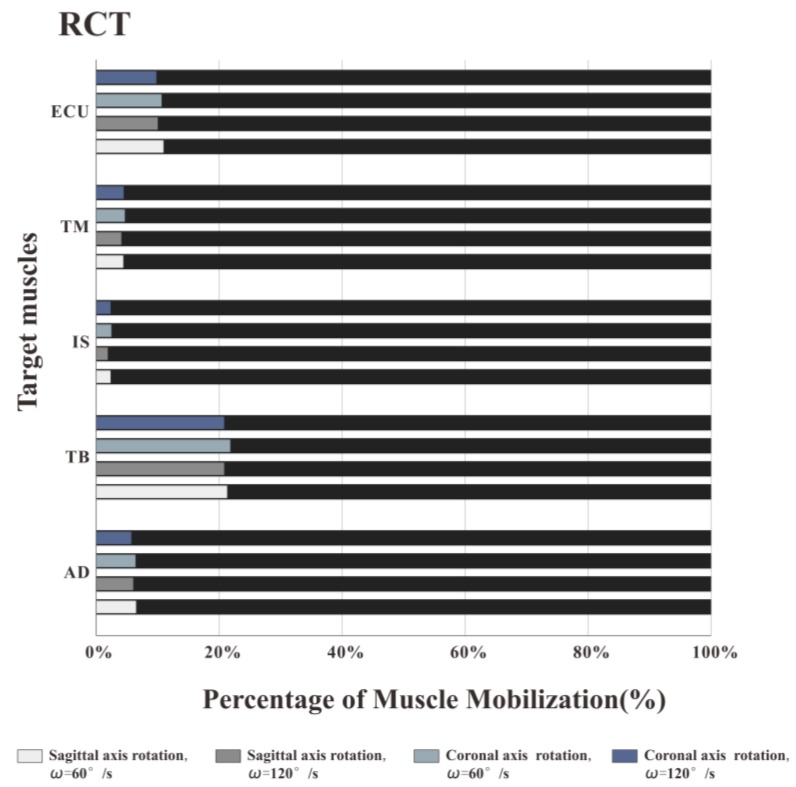
Muscle mobilization order of training by rotator cuff trainer (RCT). The order of muscle mobilization was: infraspinatus (IS), teres minor (TM), anterior deltoid (AD), extensor carpi ulnaris (ECU), and then triceps brachii (TB).

**Figure 5 healthcare-08-00024-f005:**
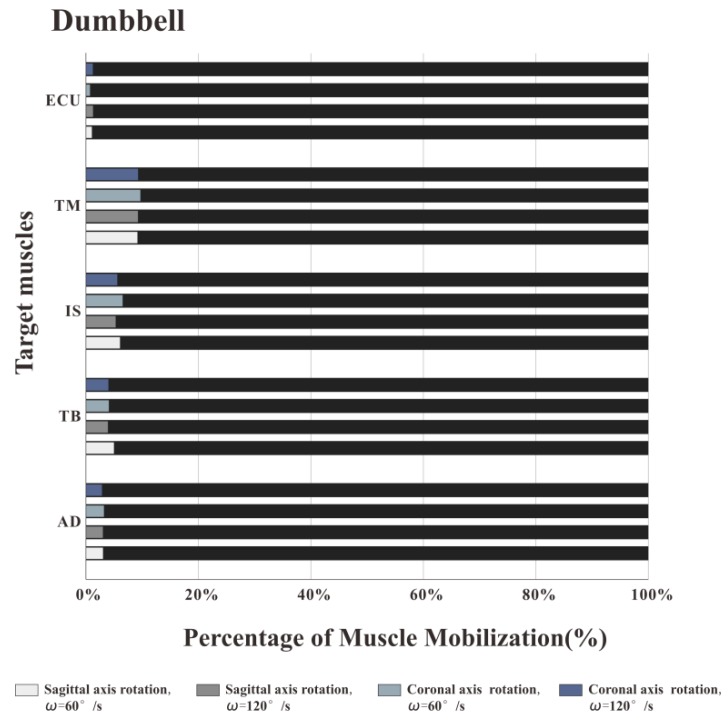
Muscle mobilization order of training by dumbbell. The order of muscle mobilization was: ECU, AD, TB, IS, and then TM.

**Figure 6 healthcare-08-00024-f006:**
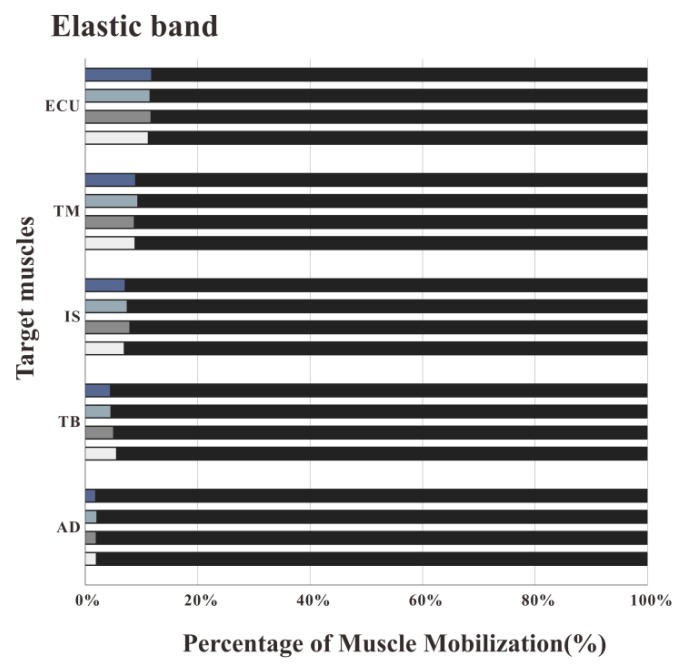
Muscle mobilization order of training by dumbbell. The order of muscle mobilization was: AD, TB, IS, TM, and then ECU.

**Figure 7 healthcare-08-00024-f007:**
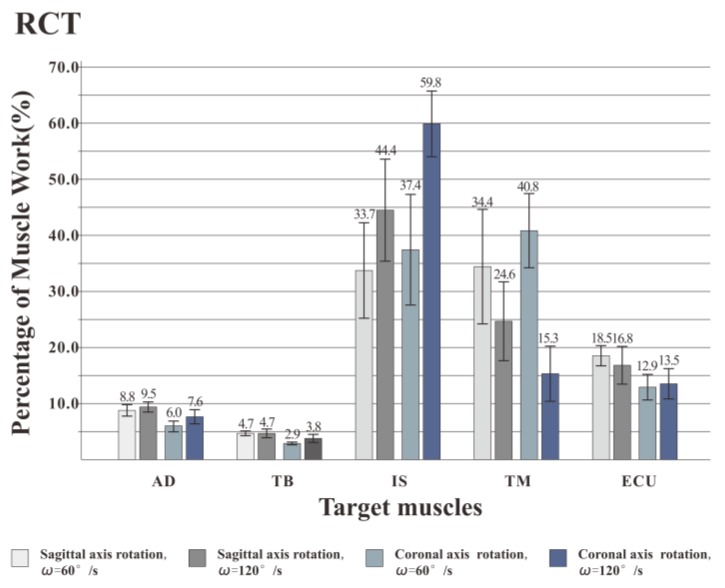
Muscle work percentage of training by RCT. The IS had the highest percentage of work, followed by TM, while TB had the least work (less than 10%). Compared with the movement speed of 60°/s, when the subjects exercised at 120°/s, the proportion of work done by IS increased significantly, while the proportion of work done by TM decreased.

**Figure 8 healthcare-08-00024-f008:**
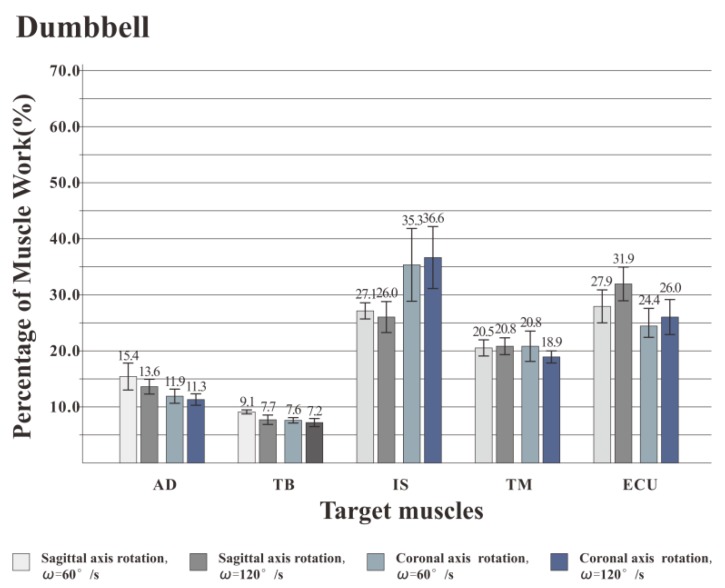
Muscle work percentage of training by dumbbell. IS was mainly used during the dumbbell task, and IS had the highest proportion of work, followed by ECU.

**Figure 9 healthcare-08-00024-f009:**
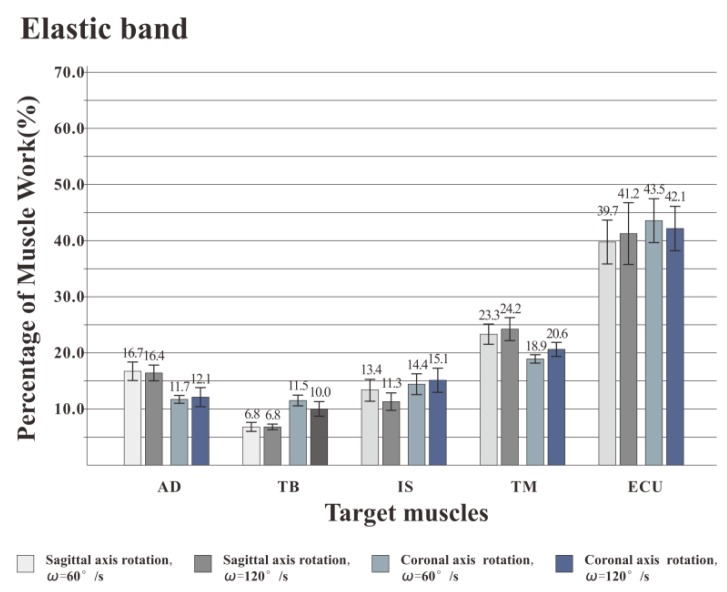
Muscle work percentage of training by elastic band. The ECU had the highest work ratio, and the TM was the second.

**Figure 10 healthcare-08-00024-f010:**
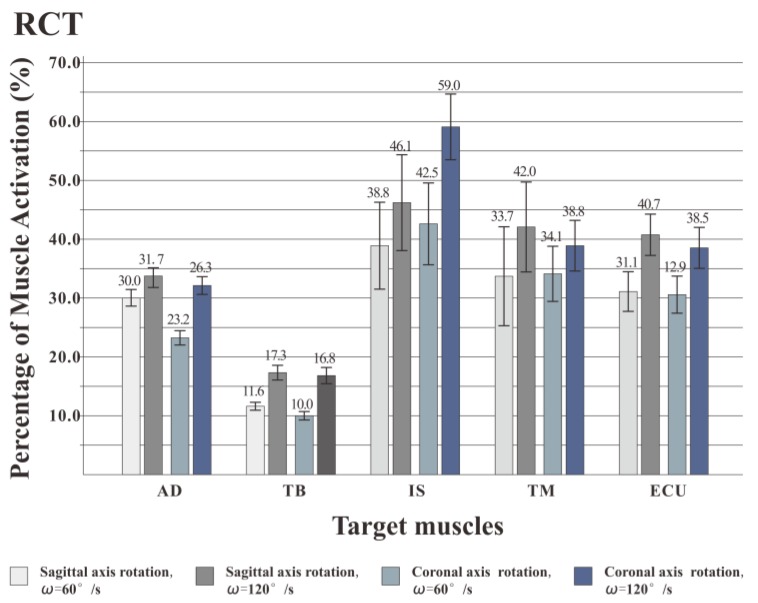
Muscle activation percentage of training by RCT. When rotated at a speed of 120°/s, the activation intensity of the muscle was significantly higher than at 60°/s. At both speeds, the IS activation intensity of rotation around the coronal axis was higher than that around the sagittal axis. The TM activation intensity in the coronal axis posture was lower than in the sagittal axis posture.

**Table 1 healthcare-08-00024-t001:** Instructions for the location of the muscle to be tested and the electrode sheet to be pasted.

Name of the Target Muscle	Infraspinatus (IS)	Teres Minor (TM)	Anterior Deltoid (AD)	Triceps Brachii (TB)	Extensor Carpi Ulnaris (ECU)
Electrode location	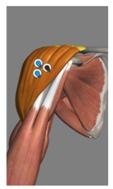	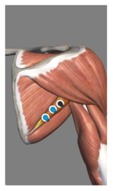	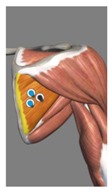	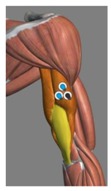	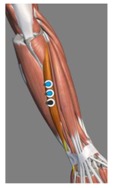

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
