# Peer review of "Evaluation of a New Rotator Cuff Trainer Based on Oscillating Hydraulic Damping"

_healthcare, 2020, doi:10.3390/healthcare8010024_

Round 1
Reviewer 1 Report
The authors compare a new rotator cuff trainer (RCT, Oscillating hydraulic damping) with Dumbbell and elastic bands, and analyze and investigate separately each muscle in the shoulder joint, so as to determine the efficacy of RCT.
The fact that muscle strengthening exercises in different shoulder rotator cuffs and individual adjacent shoulder muscles are performed is important and interesting. The Design and Results are noted in detail. The fact that the report in its entirety is very detailed makes it easy to understand. Because conditions such as rotator cuff tears and rotator cuff tendinitis occur in a considerable number of persons, I feel that this paper is of interest and has merit.
Minor points
In Figure 4-10, the addition of some explanation not only in the text but also in the Figure would make it easier to look at and understand.
Although it is stated in the text that RCT using Oscillating hydraulic damping is both safe and inexpensive (L388), it would be better to add some comment on in what way it is safe and also some comparison with other methods. Please also include some reference papers in your explanation of these points. Also, please outline the reasons why the cost is lower as compared with the two other groups(I am concerned about issues such as preparation of the machine).
In future studies it will be necessary to also consider other types of cases such as those with pre-existing rotator cuff or muscle damage(an elderly group with rotator cuff and/or muscle degeneration already present)for the sake of comparison. In the present paper as well, it would be helpful if mention could be made of any changes that were made in the implementation of the procedures according to (advanced) age, as well as any special concerns that arose related to age.
Author Response
Dear reviewer:
Thank you very much for your valuable comments about our paper submitted to the healthcare-698700.
We thank you for your encouraging and professional review of our previous manuscript. We have carefully considered the comments and have revised the manuscript accordingly. The relevant problems had been revised in the original manuscript according to your comments. We also responded point by point to each comment as listed below. The bold word below is our modification in manuscript.
In Figure 4-10, the addition of some explanation not only in the text but also in the Figure would make it easier to look at and understand.
Response: Thank you for your professional advice. We have added the explanation in the Figure4-10 title. This is a common way in academic papers. We believe it will make our Figures be easier to look and understand.
Although it is stated in the text that RCT using Oscillating hydraulic damping is both safe and inexpensive (L388), it would be better to add some comment on in what way it is safe and also some comparison with other methods. Please also include some reference papers in your explanation of these points. Also, please outline the reasons why the cost is lower as compared with the two other groups(I am concerned about issues such as preparation of the machine).
Response: Thank you for your professional suggestions. The RCT using oscillating hydraulic damping is more safer than ordinary gym equipment. There are three reasons we added in our paper.
Firstly, oscillating hydraulic damping device is safe for exercise because of no inertia. Shoulder joint is a unstable active joint that it will be easy to be injured by a sudden external force. And no inertia has proven to be very important in strength training especially for rotator cuff (Line 104-106). We add the reference [16], [17] and [33].
Secondly, the subject could also set the fixed angle of the RCT. The adjustment range was 0-90°, which can adapt to different shoulder rotation movements and exercise angles for the healthy or injured ones depending on individual condition (Line 120-122).
Thirdly, the RCT is equipped with with elbow support (Line 345-346). The elbow support will help to keep the shoulder rotation movement smooth and uninterrupted. What’s more, it will help the patients with pre-existing rotator cuff or muscle damage to complete the training in a safe way, because the muscles may not have enough strength to overcome the gravity of their arm (Line 351-353).
Of course, the RCT is more expensive than dumbbell and elastic band. The dumbbell and elastic band are common equipment to strengthen our rotator cuff because people have no other economical and effective means for shoulder training. We added some description about this point: The the oscillating hydraulic damping device is a developed hydraulic device which is widely used in mechanical equipment. Compared with other large-scale fitness equipment, the manufacturing cost of the RCT is low and it can be used in gym and rehabilitation center in the future if it has good fitness effect (Line 123-126).
3. In future studies it will be necessary to also consider other types of cases such as those with pre-existing rotator cuff or muscle damage(an elderly group with rotator cuff and/or muscle degeneration already present)for the sake of comparison. In the present paper as well, it would be helpful if mention could be made of any changes that were made in the implementation of the procedures according to (advanced) age, as well as any special concerns that arose related to age.
Response:Thank you for your professional suggestion on our future work! During the RCT design,we had considered the other types of cases such as those with pre-existing rotator cuff or muscle damage (an elderly group with rotator cuff and/or muscle degeneration already present). So,in order to meet the need for shoulder rotation training of light resistance, medium resistance, and heavy resistance, a 12-gear knob was designed to set the exercise resistance. Each level of resistance is divided into four levels. Subjects with pre-existing rotator cuff or muscle damage can select the small resistance depending on individual condition. Moreover, the subject could also set the fixed angle of the RCT. The adjustment range was 0-90°, which can adapt to different shoulder rotation movements and exercise angles for the healthy or injured ones. After setting, the subject performed reciprocating shoulder rotation training(Line116-122). It is a great pity that we had not tested the patients,but in the future studies,And we will increase the test of deep rotator cuff muscles (scapular and supraspinatus muscles) to explore the effect of different postures and speeds on the absolute strength, endurance and explosive force of rotator cuff muscles. Then, according to different individuals, the personalized exercise prescription of rotator cuff strength training can be established(Line435-440).
Overall, thank you very much for your valuable comments, which made this paper better. Hope these will make it more acceptable for publication. If you have any question about this paper, please do not hesitate to let me know. Thank you!
Sincerely yours,
Jianfeng Wu
Reviewer 2 Report
The authors presented an interesting study to design an innovative strength trainer specifically for shoulder rotator cuff based on oscillating hydraulic damping. The feasibility and validity of the RCT were explored and tested.
To this reviewer, the topic is interesting and the manuscript was well-written. However there are some minor limitations. The authors need to address them before publishing their interesting study.
I believe that the word “Ergonomic” is redundant and irrelevant in the title. Please remove it. The authors need to justify why they use these five target muscles. According to the results (Figures 7-10), different speeds have inconsistent effects on different muscles. The authors should explain these differences. It seems that Dumbbell and Elastic band works better for AD and TB muscles than RCT in figures 7-9. The authors should explain it. All of the subjects were males. So the authors may not generalize the results for females. They need to mention this limitation in discussion. They tested the RCT for healthy subjects. It is not clear what the effects of this device for patients. In the future study, this device should be evaluated for clinical applications. This limitation should be mentioned in discussion section.Author Response
Dear reviewer:
Thank you very much for your valuable comments about our paper submitted to the healthcare-698700.
We thank you for your encouraging and professional review of our previous manuscript. We have carefully considered the comments and have revised the manuscript accordingly. The relevant problems had been revised in the original manuscript according to your comments. We also responded point by point to each comment as listed below. The bold word below is our modification in the original manuscript.
I believe that the word “Ergonomic” is redundant and irrelevant in the title. Please remove it.
Response: Thank you for your professional advice. We have removed the word “Ergonomic”(LINE 2).
The authors need to justify why they use these five target muscles.
Response: Thank you for your professional suggestions. We have added the explanation in LINE172-179: In the rotator cuff muscle groups, the supraspinatus and scapular are double muscles, and surface EMG cannot be measured. In this experiment, the infraspinatus (IS) and teres minor (TM) were selected to represent the rotator cuff muscle group for subsequent evaluation. Also, several muscles with different functions related to the rotation of the shoulder were selected as the target muscles: specifically, deltoid anterior (AD), triceps brachii(TB), and extensor carpi ulnaris (ECU) . The contraction of AD drives internal rotation of the shoulder joint. The ECU is an auxiliary muscle in overcoming the resistance of rotation, and the TB helps to maintain the position in movement of shoulder rotation exercise.
According to the results (Figures 7-10), different speeds have inconsistent effects on different muscles. The authors should explain these differences.It seems that Dumbbell and Elastic band works better for AD and TB muscles than RCT in figures 7-9. The authors should explain it.
Response:Thank you for your professional suggestion! We have added the explanation in LINE388-397 and LINE407-411:
From the Figure 7-9, we also found that different speeds have inconsistent effects on different muscles. IS is one of the most powerful rotator muscles of the shoulder, and TM is an auxiliary muscle for shoulder rotation[35]. When using the RCT, with the increase of the speed, the IS force output increases quickly to maintain a faster speed. Unlike IS, the increase of TM force output is far less than rotator cuff muscles. As a result, the percentage of muscle work of IS increased with a higher speed, and the percentage of muscle work of TM decreased. When using dumbbell or elastic band, the AD and TB contributed a lot to keep the position of movement which led to a better work for AD and TB muscles than RCT. All tested muscles might be almost equally important to the shoulder exercise that the the percentage of muscle work had no significant change. ( Line388-397)
As the rhythm of the movement increases, the muscle contraction level increases, the amount of discharge increases, and the muscle force output increases to maintain a faster movement speed. Therefore, in the experiment, when the shoulder rotation training is performed at a higher speed (120°/s), the activation intensity of the muscle is stronger. ( Line407-411)
3. All of the subjects were males. So the authors may not generalize the results for females. They need to mention this limitation in discussion.They tested the RCT for healthy subjects. It is not clear what the effects of this device for patients. In the future study, this device should be evaluated for clinical applications. This limitation should be mentioned in discussion section.
Response:Thank you for your professional suggestion on our future work! During the RCT design,we had considered the other types of cases such as those with pre-existing rotator cuff or muscle damage(an elderly group with rotator cuff and/or muscle degeneration already present). So,In order to meet the need for shoulder rotation training of light resistance, medium resistance, and heavy resistance, a 12-gear knob was designed to set the exercise resistance. Each level of resistance is divided into four levels. Subjects with pre-existing rotator cuff or muscle damage can select the small resistance depending on individual condition. Moreover, the subject could also set the fixed angle of the RCT. The adjustment range was 0-90°[19], which can adapt to different shoulder rotation movements and exercise angles for the healthy or injured ones. After setting, the subject performed reciprocating shoulder rotation training(Line116-122). It is a great pity that we had not tested the patients,but in the future studies,we will add the test for female and patients, to verify the feasibility and validity of RCT to different people. And then increase the test of deep rotator cuff muscles (scapular and supraspinatus muscles) to explore the effect of different postures and speeds on the absolute strength, endurance and explosive force of rotator cuff muscles. Then, according to different individuals, the personalized exercise prescription of rotator cuff strength training can be established(Line435-440).
Overall, thank you very much for your valuable comments, which made this paper better. Hope these will make it more acceptable for publication. If you have any question about this paper, please do not hesitate to let me know. Thank you!
Sincerely yours,
Jianfeng Wu
